# Untangling the Causal Links between Satellite Vegetation Products and Environmental Drivers on a Global Scale by the Granger Causality Method

Dávid D. Kovács [1,*] , Eatidal Amin [1] , Katja Berger [1,2] , Pablo Reyes-Muñoz [1] and Jochem Verrelst [1]

1   Image Processing Laboratory (IPL), Universitat de València, C/Catedrático José Beltrán, 2,
    46980 Paterna, Spain; eatidal.amin@uv.es (E.A.); katja.berger@uv.es (K.B.); pablo.reyes@uv.es (P.R.-M.);
    jochem.verrelst@uv.es (J.V.)
2   Mantle Labs GmbH Grünentorgasse 19/4, 1090 Vienna, Austria
*   Correspondence: david.kovacs@uv.es; Tel.: +34-623-214-755

**Abstract:** The Granger Causality (GC) statistical test explores the causal relationships between different time series variables. By employing the GC method, the underlying causal links between environmental drivers and global vegetation properties can be untangled, which opens possibilities to forecast the increasing strain on ecosystems by droughts, global warming, and climate change. This study aimed to quantify the spatial distribution of four distinct satellite vegetation products' (VPs) sensitivities to four environmental land variables (ELVs) at the global scale given the GC method. The GC analysis assessed the spatially explicit response of the VPs: (i) the fraction of absorbed photosynthetically active radiation (FAPAR), (ii) the leaf area index (LAI), (iii) solar-induced fluorescence (SIF), and, finally, (iv) the normalized difference vegetation index (NDVI) to the ELVs. These ELVs can be categorized as water availability assessing root zone soil moisture (SM) and accumulated precipitation (P), as well as, energy availability considering the effect of air temperature (T) and solar shortwave (R) radiation. The results indicate SM and P are key drivers, particularly causing changes in the LAI. SM alone accounts for 43%, while P accounts for 41%, of the explicitly caused areas over arid biomes. SM further significantly influences the LAI at northern latitudes, covering 44% of cold and 50% of polar biome areas. These areas exhibit a predominant response to R, which is a possible trigger for snowmelt, showing more than 40% caused by both cold and polar biomes for all VPs. Finally, T's causality is evenly distributed amongst all biomes with fractional covers between ~10 and 20%. By using the GC method, the analysis presents a novel way to monitor the planet's ecosystem, based on solely two years as input data, with four VPs acquired by the synergy of Sentinel-3 (S3) and 5P (S5P) satellite data streams. The findings indicated unique, biome-specific responses of vegetation to distinct environmental drivers.

**Keywords:** vegetation sensitivity; Google Earth Engine; Granger Causality; FAPAR; LAI; SIF; NDVI; ERA5; environmental stress; causality

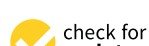



## 1. Introduction

Our planet faces global warming and climate change-induced hydro-meteorological challenges that ought to be mitigated [1]. Droughts and extreme heat waves pose a great threat to global vegetation health [2]. Hence, gaining a deeper understanding of the correlations among different climatological, environmental, and biophysical factors has become increasingly crucial [3]. Droughts limit optimal vegetation growth, disturb water relations, and impair water use efficiency in plants [4], and, as these natural disasters are more frequent, it is an urgent issue to monitor the global ecosystem's health [5]. To understand the fundamental drivers of drought evolution, even the smallest changes in the environment demand spatially explicit monitoring. Abrupt increments in temperatures, and uneven temporal distributions of precipitation lead to dryer soils, ultimately causing droughts

that negatively affect the ecosystem, agriculture, and water resources [6,7]. Linking spatially explicit vegetation dynamics with climatic factors can facilitate the interpretation of ecosystem feedback to natural stresses [8].

The exploitation of optical satellite data streams opened up possibilities for effectively monitoring the onset, duration, and magnitude of droughts [9]. Throughout the literature on assessing vegetation dynamics and health by means of satellite imagery, vegetation indices were mainly investigated due to their long-term availability and wide range of applications [10].

Alternatively, more advanced metrics, such as quantitative biophysical variables or solar-induced chlorophyll fluorescence (SIF), for evaluating vegetation productivity can yield more consistent findings. While most of the light absorbed by leaves is utilized for carbon assimilation, a fraction is released as heat and emitted as SIF [11]. Quantifiable biophysical variables and SIF not only aid in deciphering the captured SIF signal to estimate photosynthetic activity but also form the essential foundation for monitoring ecosystem dynamics across the spatiotemporal domain [12].

Hence, the sensitivity of vegetation to environmental drivers, understood as the set of physical variables influencing vegetation physiology with statistical meaning, is far better assessed by investigating distinct satellite vegetation products (VP). These include biophysical variables, namely: (1) the fraction of absorbed photosynthetically active radiation (FAPAR) and (2) the leaf area index (LAI). Additionally, (3) SIF and the (4) normalized difference vegetation index (NDVI) are also considered. The targeted VPs can be described as follows:

The FAPAR quantifies the proportion of solar radiation within the 400–700 nm spectral range that is absorbed by the vegetation, relative to the total radiation absorbed at the surface [13,14]. It is an important variable, as it contributes to the monitoring of primary productivity [15]. Additionally, the FAPAR serves as an observational constraint when simulating carbon fluxes between vegetation and the atmosphere in terrestrial biosphere models [14,16].

The LAI is described as half of the total intercepting leaf area per unit of ground surface area [17]. It strongly contributes to canopy photosynthesis and evapotranspiration [15]. The LAI is considered a key component of biogeochemical cycles in ecosystems [18].

SIF, which represents the radiant flux emitted in the wavelength range from 650 to 800 nm, serves as the most direct measurable indicator of the photosynthetic processes in plants [11,19,20].

The NDVI is the most extensively used vegetation index used in Earth observation (EO) [21]. Through the computation of the difference between bands that measure red and near-infrared (NIR) reflectance, the NDVI highlights the distinctive signatures of green vegetation [22].

The understanding of global vegetation anomalies lies within unraveling its causal dependencies from abiotic factors, such as fluctuations in soil moisture content and precipitation (i.e., water availability), as well as changes in air temperature and solar radiation (i.e., energy availability). Further, the water availability to plants drives changes in stomatal guard cell turgor pressure, which actively regulates the stomatal opening. Thus, greater supplies of water will increase stomatal opening, leading to augmented rates of $CO_2$ uptake for photosynthetic activity [23,24]. The causality analysis of VPs to precipitation (P) and soil moisture (SM) may possibly explain anomalies in vegetation dynamics associated with water limitation. Additionally, the health of vegetation across the Earth's surface hinges on the environmental temperature and its fluctuations. Terrestrial vegetation undergoes growth, reproduction, and productivity patterns related, among others, to environmental temperature [25]. Moreover, through various factors including the amount of total energy, photons, spectral characteristics, duration, and photoperiod, light also governs numerous aspects of plant development [26]. Therefore, apart from water, plants' responses to air temperature (T) and solar radiation (R) also play a crucial role in photosynthetic activity [27]. The spatial patterns of vegetation type and function vary in connection with the existing environmental circumstances. Climatic and soil-related conditions are known to

directly influence different plant species distributions. To model vegetation anomalies to environmental stress (e.g., water and energy availability), the analysis needs to be adapted to account for the role of dominant biomes in determining how vegetation responds to environmental drivers and disturbances [28].

Vegetation anomalies and responses to multiple ELVs have previously been monitored with multi-decadal data (e.g., [29–31]). A variety of approaches have previously been applied to investigate vegetation–environment interconnections. For instance, Chen et al. [29] used deep learning methods to simulate the long-term vegetation greenness dynamics and assess their sensitivity, whilst Telesca et al. [31] implemented the Fisher–Shannon statistical algorithm to assess the complexity and non-stationarity of non-linear time series. Bao et al. [30] used regression analysis on long-term NDVI data to assess the correlation between vegetation and environment. Yet, there is a lack of shorter-term sensitivity analysis to assess the environmental stress on ecosystem integrity. Although efforts have been presented to apply analysis to shorter timescales [32,33], these studies usually focused on a regional scale and not the globe as a whole. For example, Olthof and Latifovic [32] investigated the response of the NDVI to the preceding 10 to 40 day temperature anomalies over Canada, while Zhang et al. [33] applied monthly relative anomalies to correlate the temporal dynamics of Gross Primary Production (GPP), SIF, and absorbed photosynthetically active radiation (APAR) to investigate the vegetation's response to droughts in Victoria, Australia.

While the usage of simple correlation and multi-linear regression analyses have significantly contributed to the comprehension of the interconnections between climate and vegetation, these are generally inadequate to reveal causalities [34]. Therefore, an important step towards understanding the stress-response relationship between environment and vegetation is the exploration of causal relationships, such as the Granger Causality (GC) [35]. The GC is a statistical concept used to assess whether one time series can be considered a predictor for another time series. If past values of one time series provide information that helps predict the second time series, then the first time series is causal to the second one [36]. The GC methodology has previously been introduced in economics and is widely used throughout neuroscience, finance, and genomics [37–39], allowing for the assessment of causal influence between two time series variables. Applying this method to global-scale vegetation and environmental data helps uncover the directional relationship between stress factors and vegetation responses.

The understanding of the biome's reactions to forthcoming long-term climate shifts involves assessing the sensitivity of worldwide ecosystems on shorter timescales. Conveniently, the ever-increasing satellite imagery data streams and reanalysis data on the atmosphere, biosphere, and hydrosphere, provide means for interpreting causal connections with novel, data-driven approaches, such as the GC. Previous research made efforts to quantify vegetation sensitivity to environmental variables with the GC method on global [34] and regional scales [40,41] using the NDVI or LAI. However, the scientific literature still lacks the comprehensive, planetary-scaled, multiple VP analysis by means of the GC. Consequently, this study is determined to untangle how multiple ELVs cause vegetation to react to stress across the globe by applying the GC method. The main objective of this study, therefore, is to quantify, map, and evaluate the sensitivity of satellite vegetation products to abiotic environmental factors from water and energy availability at the global scale. The presented analysis addresses the application of satellite data streams and GC analysis in understanding the complex interactions between the environment and vegetation dynamics.

## 2. Materials and Methods

To successfully investigate the global GC, the following steps were implemented into our analysis; see Figure 1. First, the main ELVs were identified, as possible causal factors on four main VPs. These were all adequately gathered and implemented into Google Earth Engine (GEE) to smooth processing on a single platform. Second, these products were

accessed by linking GEE to Python via the REST API. Using Python's robust statistical capabilities, ELV-caused VP temporal profiles were identified to calculate, spatially explicit, pixel-wise Granger Causality metrics. Ultimately, global maps for GC metrics were retrieved and statistical analysis was conducted on five dominating biomes. Conclusions were drawn as to how different environmental drivers affect and cause changes in the products and properties of vegetation over different biomes. Hereinafter, the ELV→VP notation will be used to indicate different variables' causal relationships.

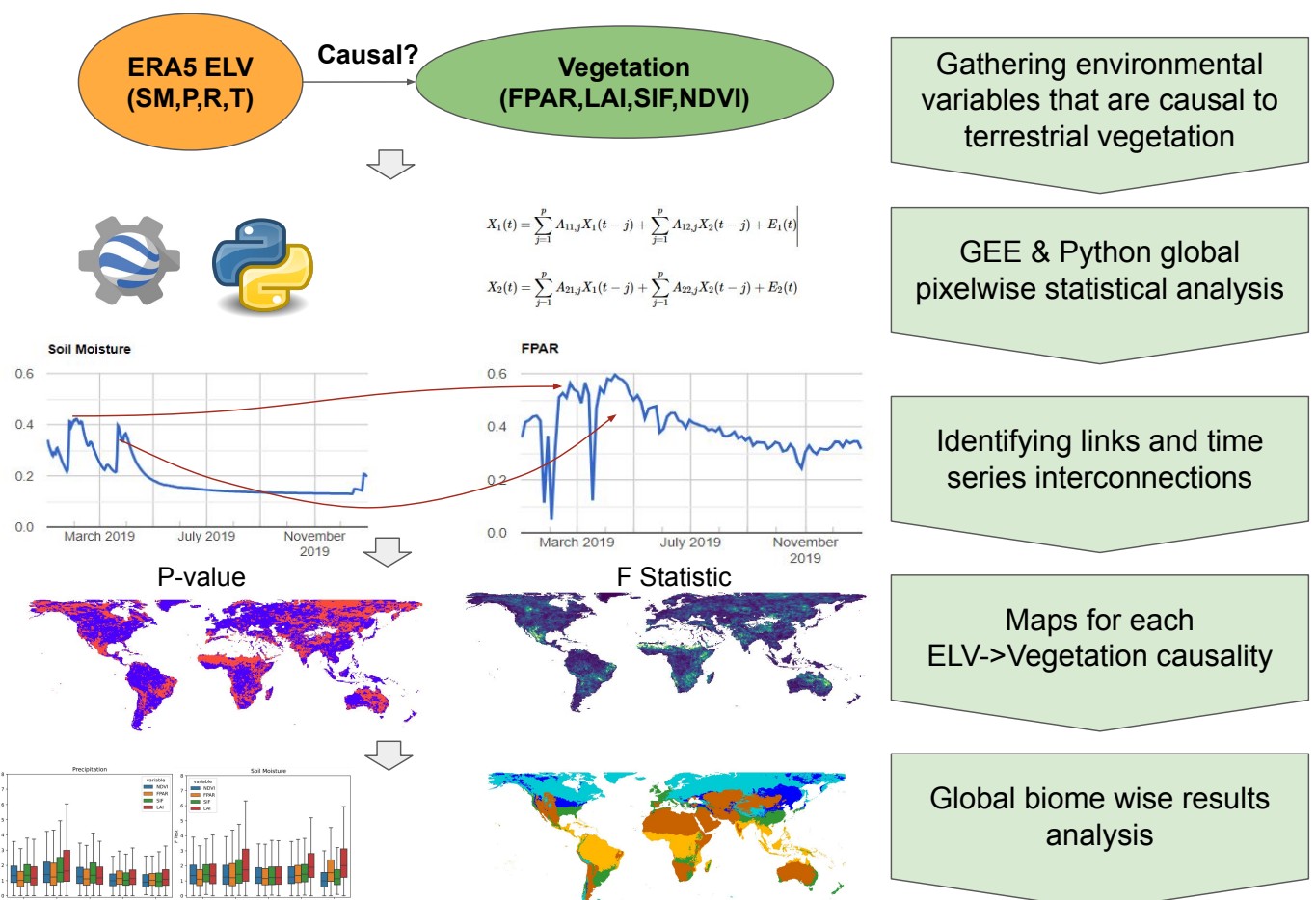

**Figure 1.** Flowchart analysis followed throughout this study.

### 2.1. Granger Causality

To apply the GC method, suppose we have two time series: $x = [x1, x2, ..., xN]$ and $y = [y1, y2, ..., yN]$, where $N$ represents the length of the time series in days. Consider $x$ the vegetation variable and $y$ the environmental surface variable. For Granger's predictive causality, the goal is to predict the value of $y(t)$ at a specific timestamp $t$, based on the historical values of $x$ and $y$ in previous timestamps [34]. Granger states that the variable $x$ is Granger caused (G-Caused) by $y$, if the autoregressive forecast of $x$ is superior when additional information is provided by the variable $y$. We used a Vector autoregressive (VAR) model that considers two variables simultaneously and models the relationships between them, investigating the causal links between $x$ and $y$. The mathematical calculation is given as follows:

$$X(t) = \sum_{j=1}^{p} A_{xx,j} X(t-j) + \sum_{j=1}^{p} A_{xy,j} Y(t-j) + E_1(t) \tag{1}$$

$$Y(t) = \sum_{j=1}^{p} A_{yx,j} X(t-j) + \sum_{j=1}^{p} A_{yy,j} Y(t-j) + E_2(t) \tag{2}$$

where $p$ is the maximum allowable lagged observations in the model, and $A$ are matrices containing the influences of each previous observation on the predicted values $X$ and $Y$ at a time $t$. $E_1$ and $E_2$ are the prediction errors. If the variance of the error $E_1$ is reduced by including $Y$ in the prediction, then it can be stated that $Y$ is G-causing $X$ [35]. The null hypothesis $H_0$, stating that $Y$ does not G-cause $X$, is rejected when the causality test is statistically significant ($p < 0.05$). In any other case, $H_0$ is not rejected, and it cannot be concluded that $X$ is only G-Caused by $Y$ [42,43]. Within VAR modeling, a lag refers to the number of previous observations of each variable that are included as explanatory variables when modeling the current value of that variable [44]. This study used 100 days as the lag values.

To enable calculating valid results, note that the two time series data streams have to be normalized to avoid seasonal effects. This will remove the trends in the time series that are inherent with the changes in seasons (e.g., the increment of temperature towards summer, decrease in the LAI throughout the winter months). Thus, the GC calculations will solely be performed with the normalized temporal profiles; they only consider the (minor) fluctuations in both the ELV and VP and purely consider the causality from the changes in one to another, avoiding the larger, seasonal changes. Throughout the presented analysis, the first discrete difference normalization method was used. This method was selected due to its simplicity and ease of application in the processing algorithm. This normalization technique obtains each time series value by using the following logic:

$$y(t) = p(t) - p(t-1) \tag{3}$$

where $y$ is the normalized time series value at a time $t$. $p(t)$ is the original time series value, and $p(t-1)$ is the original value one timestep before $p(t)$. Thus, the series of the differences, $p(t) - p(t-1)$, for both the ELV and VP will be used to calculate the GC.

The F statistic (F stat) of the Granger Causality analysis is obtained from the F-test, which is computed as the ratio between two summed squared residuals from Equations (1) and (2) ($RSS_x$ and $RSS_y$). It is calculated as follows, with $n$ data points and $(p_X - p_Y, n - p_Y)$ degrees of freedom:

$$F = \frac{\left(\frac{RSS_X - RSS_Y}{p_X - p_Y}\right)}{\left(\frac{RSS_Y}{n - p_Y}\right)} \tag{4}$$

As is also used by Wismüller et al. [45], the hereby presented analysis will take into account the F statistic, to evaluate the GC links between different variables. Throughout Section 3.1, global maps representing the F-statistics are shown for various vegetation–environment causality links. Throughout the analysis, the direction of the GC was only investigated on vegetation from the environment (ELV→VP). The reversed case was not considered, (VP→ELV), although ideally the probability of the reversed case ought to also be ascertained.

### 2.2. Vegetation Products and Environmental Land Variables

The investigated vegetation product datasets, i.e., the FAPAR, the LAI, SIF, and the NDVI, were derived and captured by the PROBA-V, Sentinel-3 (S3), and Sentinel-5P (S5P) satellites. Two years of input data (January 2020–December 2021) was used with 10-day composites. This temporal window was selected because it proved to be the common temporal domain for all datasets whilst still maintaining a relatively short timescale.

The LAI, FAPAR, and NDVI products were used from the Global Land Service (CGLS) datasets, originating from PROBA-V and Sentinel-3 OLCI. The CGLS datasets were downloaded from https://land.copernicus.eu/global/themes/vegetation (accessed on 8 August 2023). The retrieval of the FAPAR and LAI involved the application of neural networks on Top-of-Canopy (TOC) input reflectances in red, near-infrared, and shortwave infrared to infer the estimates [46]. The NDVI was calculated using atmospherically and angular corrected Red and NIR TOC reflectances [47].

Amongst many atmospheric satellite missions providing SIF data, S5P TROPOMI (TROPO-spheric Monitoring Instrument) offers the best trade-off between spatial and temporal resolutions [48]. The baseline TROPOMI SIF (TROPOSIF) retrievals are inferred using the 743–758 nm range. Additionally, a secondary SIF dataset obtained from an extended fitting window (735–758 nm window) is incorporated. The SIF data were obtained by directly downloading them from this website: http://ftp.sron.nl/open-access-data-2/TROPOMI/tropomi/sif/v2.1/l2b/ (accessed on 7 July 2023). The data are provided in the form of ungridded L2B daily data. SIF is derived through a data-driven approach using top-of-atmosphere (TOA) radiance data in the far-red spectral region collected by TROPOMI [49]. The dataset has a spatial resolution of 7 × 3.5 km and daily revisit time. Daily SIF data were used to create the 10-day temporal composites. The summarized description of the VPs can be found in Table 1.

**Table 1.** Vegetation products and their retrieval methods used throughout this study.

| Analysis Product | Spatial Resolution | Temporal Granularity | Algorithm/Retrieval Approach | Sensor | Unit |
|---|---|---|---|---|---|
| LAI/FAPAR | 300 m | 10 Day | Neural networks trained with reflectance data | S-3 OLCI /PROBA-V | LAI: $(m^2/m^2)$/FAPAR: $(-)$ |
| NDVI | 300 m | 10 Day | BRDF-normalized, atmospherically corrected reflectances Further corrections for Sun-sensor geometry differences | S-3 OLCI /PROBA-V | $(-)$ |
| TROPOMI SIF(TROPOSIF) | 7 × 3.5 km | Daily | Infilling of Fraunhofer lines at 743–758 and 735–758 nm with fluorescence radiance | S5P | mW m$^{-2}$ sr$^{-1}$ |

The European Centre for Medium-Range Weather Forecasts (ECMWF) is currently producing an improved global dataset for the land component of the fifth generation of the European ReAnalysis (ERA5). ERA-5-LAND utilizes physical equations and assimilates observations to estimate atmospheric and land-surface variables. The data, available globally, are provided at a horizontal resolution of 0.1° × 0.1° in hourly, daily, or monthly aggregates [50]. The same temporal window (January 2020–December 2021) was selected for the ELVs. For assessing environmental effects on vegetation mapping, we utilized the daily data that was already available within the GEE platform, and 10-day temporal composites were created. The description [51,52] of each ELV can be found in Table 2.

**Table 2.** ERA-5-LAND variables and their definition used throughout the analysis.

| Surface Variable | Definition | Unit |
|---|---|---|
| Soil Moisture (SM) | Volume of water in soil layer 2 (7–28 cm) of the ECMWF Integrated Forecasting System. | 1 (volume fraction) |
| Precipitation (P) | Total daily precipitation sum. Accumulated liquid and frozen water, including rain and snow, that falls to the Earth's surface. | meter (m) |
| Temperature (T) | Temperature of air at 2 m above the surface. 2 m temperature is calculated by interpolating between the lowest model level and the Earth's surface, taking into account the atmospheric conditions. | Kelvin (K) |
| Shortwave solar radiation (R) | Amount of accumulated shortwave solar radiation (0.2–4 µm direct and diffuse) reaching the surface of the Earth. and the Earth's surface, taking into account the atmospheric conditions. | J/m$^2$ |

The temporal window for the analysis was 2 years, January 2020–December 2021. This temporal window was selected after conducting Granger analyses with one, two, and three years of data. The goal of this experiment was to derive the shortest timespan while achieving consistent findings. The two-year timespan led to significant improvements as opposed to the one-year window, i.e., the GC *p*-value pixels increased nearly two-fold. However, the change was not that significant when comparing the three-year timespan to the two-year timespan, i.e., the GC *p*-value pixels only increased approximately 5–10%. Similar increments were observed amongst all ELV→VP cases; exemplary GC *p*-value pixel counts are shown in Table 3 for P→LAI. Note how for a year-long or less than a year timespan, the number of GC significant *p*-value pixels is rather scarce compared to two-year observations. The GC *p*-value maps are shown for the three considered time windows in Figure A1.

**Table 3.** Exemplary GC *p*-value pixels that explicitly show causality. P→LAI.

| Time Window | 1 Year | 2 Year | 3 Year |
|---|---|---|---|
| GC significant *p*-value pixels | 4094 | 7713 | 8243 |
| Increment compared to previous year | - | 88% | 7% |

### 2.3. Global Analysis with Google Earth Engine and Python Geospatial Libraries

GEE is a highly scalable cloud computing platform tailored for geospatial analysis on planetary scales, harnessing access to a multi-petabyte dataset of EO products. It includes an interactive application programming interface (API) that enables users to easily handle and visualize results in the interactive development environment (IDE). Users can perform spatial calculations with access to the database and make requests via the JavaScript or Python APIs that are sent to Google via JSON objects by the client [53]. This investigation greatly benefited from the availability of various datasets on the platform, e.g., ERA5 Land surface variables and the access to these products in the cloud, saved cumbersome programming and computational work. Furthermore, additional TROPOSIF and CGLS products were uploaded and consequently handled in GEE. To aid batch processing and to harvest the complex computational libraries of Python, GEE's data were accessed via the REST API (Representational State Transfer Application Programming Interface) and converted into an *array* format. The global GC maps, along with their error metrics, were calculated as *arrays* before conversion back into GEE's *image* type for further processing within the platform. Both ELVs and VPs were resampled to the same (50 km) spatial resolution. This resolution allows for reduced memory usage and faster processing (around 1 h for an ELV→VP GC map processed with 2vCPU @ 2.2 GHz an 13 GB RAM). Consequently, this allowed for a global biome scale analysis while still being computationally feasible. To assess the five biomes' response characteristics to ELVs, the 50 km distribution proved to be a robust trade-off between accurate assessment and computational efficiency. The coarse resolution is further justified given that various datasets operate at a coarse spatial footprint (ERA5 11 km, TROPOSIF 7 km). For visualization purposes, additional regional analyses at an 11 km (nominal ERA5) resolution are shown in Figure A2. Processing the globe at this resolution would have taken up to a roughly ∼20-fold increase in computing time.

### 2.4. Biome-Specific Analysis

To evaluate the role of environmental variables driving distinct vegetation dynamics that are linked to varying climates, the Köppen–Geiger (KG) classification was used to calculate the F statistics per biome [54]. The global biome dataset consists of 30 specific biome categories; however, to facilitate the analysis and avoid having too many classes, this study considers five main biomes: Tropical, Arid, Temperate, Cold, and Polar. These five classes are depicted in Figure A3. Per-biome statistics were calculated and conclusions were drawn over how distinct ELV→VP links behave over varying climates. The biome classification used throughout the analysis was derived by Beck et al. [55].

## 3. Results

### 3.1. Global GC Maps

The 2-year temporal series of both ELV and VP pixels was analyzed by the GC significance test, yielding a value between 0 and 1. The resulting values were mapped and a causal/non-causal representation was obtained for each ELV→VP case. Figure 2 shows an example of a global map depicting G-Caused and Non-G-Caused pixels where the LAI is investigated on causality by soil moisture. Note that in order to ascertain whether one variable is causal to another, the GC significance test should yield a *p*-value lower than 0.05 (see also Section 2.1). Red pixels indicate causation, or *p*-values of less than or equal to 0.05, whilst blue areas show non-causation, or *p*-values of greater than 0.05. Note that the

fluctuations in soil moisture cause delayed increments in the LAI, as seen on the bottom left temporal profiles in Figure 2. Similar ELV/VP dynamics were observed over dry, arid areas where it was clearly visible that the given ELV was the causal factor for the investigated VP. Conversely, the other time series pair on the bottom right depicts a case where the dynamics of soil moisture have no influence on the time series of LAI. This does not mean that an ELV has no effect on VP, but, under the investigated circumstances, the ELV poses no direct, explicit causal effect on the vegetation dynamics. The non-causal relationship displayed in Figure 2 depicts that the temporal variations in the LAI show no major changes to the fluctuations in SM availability. Inspecting the two temporal profiles, for causal and non-causal, it can be observed that the LAI varies with a greater amplitude for non-causal, as opposed to causal, but without the direct causal link from SM, as SM stays constantly low throughout the investigated timeframe. Conversely, for the causal time series, the LAI stays low all year round, with the only variations directly related to the increments in SM. The non-causal time series did not improve when adding an additional year to the investigated timeframe. Furthermore, the GC statistics were calculated by the normalized values of these time series, as the ones depicted in Figure 2 stand for a visual representation.

Since the *p*-value represents a Boolean, True-or-False, approach to causality, the F test was used to further quantify and investigate causality, and the F statistics for GC pixels were calculated as shown in Figure 3.

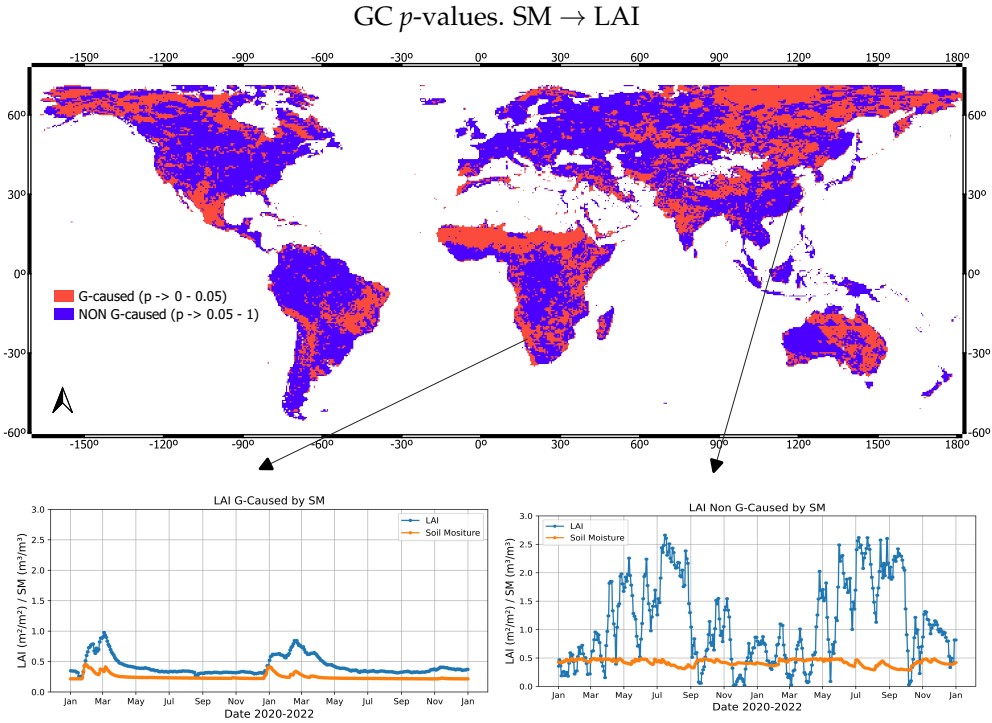

**Figure 2.** Two exemplary pairs of time series showing causal/non-causal cases of GC: SM→LAI, G-Caused *p*-values.

As can be derived from Figure 3, water availability (SM and P) proved to be an outstanding causal factor in arid biomes for all variables, with higher F statistics (>6 F stat). The most noticeable regions of strong water dependency are present over the African Savannas and Northern Australia. Inspecting Figure 3 more closely, especially the SM → LAI and P→LAI F statistic maps, one can observe the hot spots over the Great Plains in Northern America, too. Furthermore, P also causes areas over the Iberian Peninsula and arid regions near the Thar Desert. Soil moisture causality plays a prominent role in Northern Russia on polar biomes. The most significant effect of SM is observed for the LAI, over the Northernmost latitudes both for the Asian and American continents.

Areas that are driven by R are also distributed over Scandinavia and Northern Russia with minor spots in Southern Africa. Temperature limitation does not show any well-defined spatial pattern for any biomes or ELV→VP links but presents an even distribution along the globe across all climate zones.

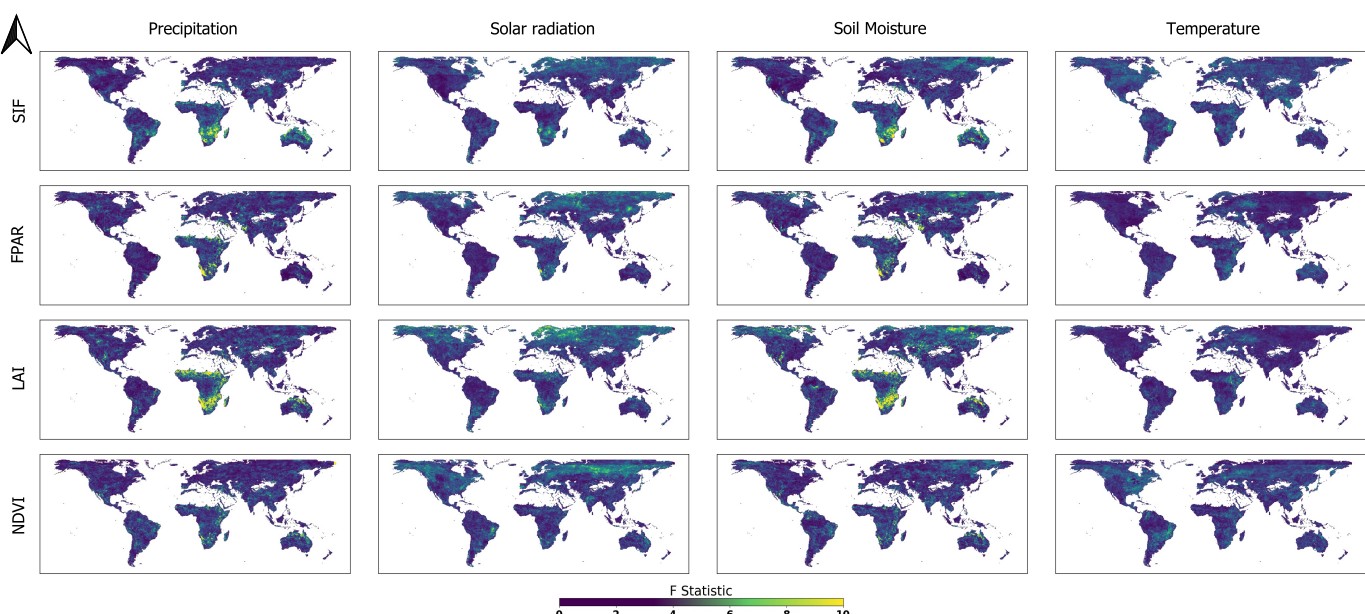

**Figure 3.** Global distribution of F statistic for each ELV →VP link.

### 3.2. Köppen–Geiger Biome Specific Analysis

Global *p*-value significance maps were calculated for the ELV→VP GC link. One exemplary *p*-value map is shown in Figure 2. Here, each of the five main biomes was analyzed to ascertain what fraction of the biome's area is susceptible to GC causality. These fractions are shown in Figure 4.

SM→LAI connections show a strong presence (>0.4) for arid, cold, and polar biomes. Moderate cover fractions describe other links, although SM →NDVI causalities are relatively scarce (0.15) on polar biomes as compared to other VPs (>0.3). P, as expected, causes larger arid areas (>0.26) for all variables, with the largest cover fraction for P→LAI (0.4). As compared to SM, P shows less effect on VPs over cold and polar climates, with all cover fractions being lower than 0.2. T shows low (∼0.15–0.2), but constant, causal distribution amongst many VPs, with higher fractions for the NDVI in cold and temperate biomes (0.26 and 0.28). R-caused VPs cover nearly half of the cold and polar biomes, with moderate fractions over the other three climate zones. R→NDVI appears to be less causal in polar regions, being merely half of the cover fraction, as suggested by other ELV→VP links in the same setting.

The results for causation links obtained in Section 3.1 were further analyzed by dividing the globe into different biomes and investigating the statistics of the *p*-values and F statistic GC cases. Each F statistic global map was masked and the corresponding biome of interest's statistical metrics were calculated accordingly. Figure 5 shows the biome-wise distribution of the GC F statistics for each ELV→VP link. One important note to consider is the non-exclusivity of causality from different ELVs. In other words, one pixel can be caused by both, e.g., precipitation and temperature; these are further assessed by the F statistics, and the maps for the primary causal variables are shown in Figure 6. The main findings from the boxplot's statistical metrics are described as follows. Water availability exhibits the greatest influence on the LAI; SM→LAI is a prominent causal link on arid, cold, and polar biomes, and P→LAI is also a strong link on arid biomes, similar to those investigated in Figure 4. R poses a substantial causal effect for all VPs on cold and polar biomes, whereas it does not show such consistent driving forces on other biomes.

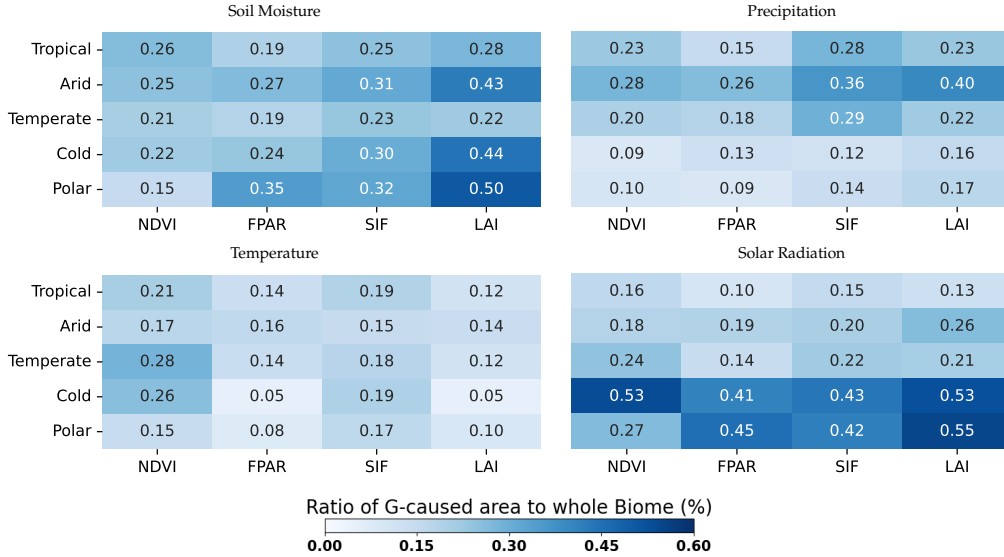

**Figure 4.** Fractions of G-Caused area of the given biome to the whole biome's area.

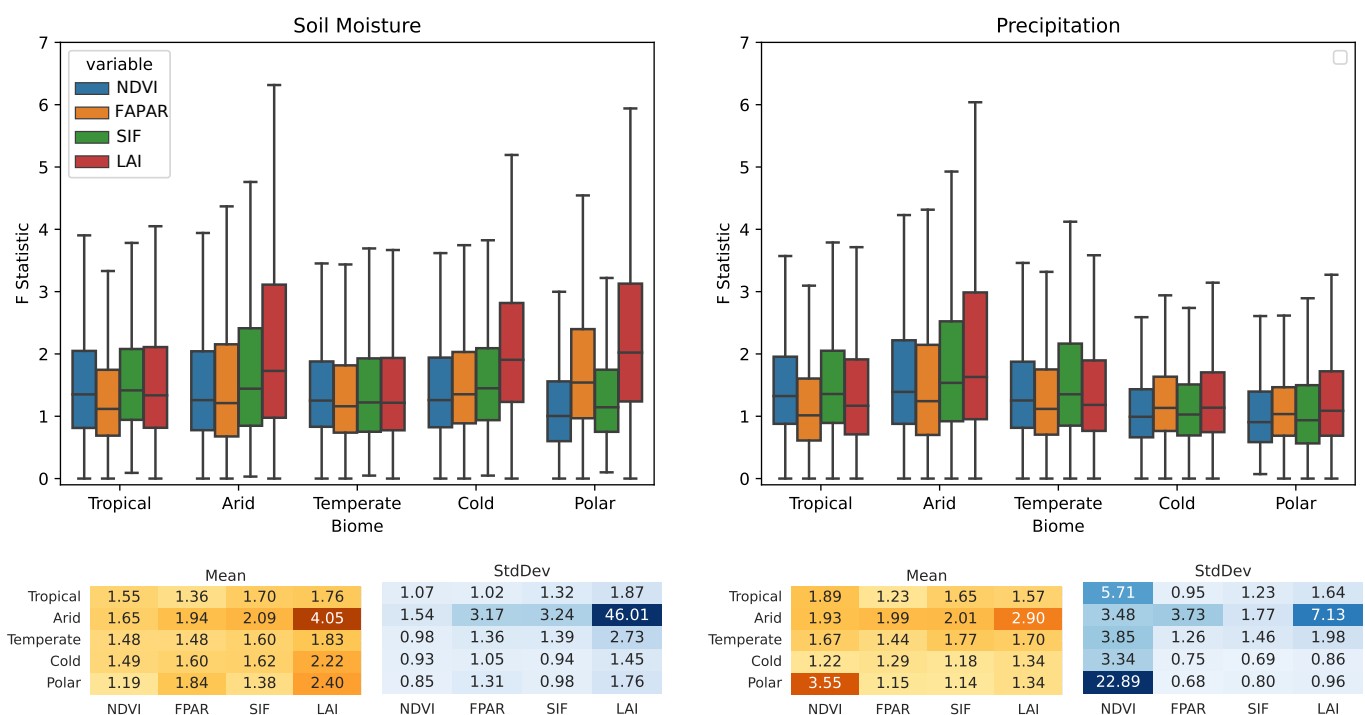

**Figure 5.** *Cont.*

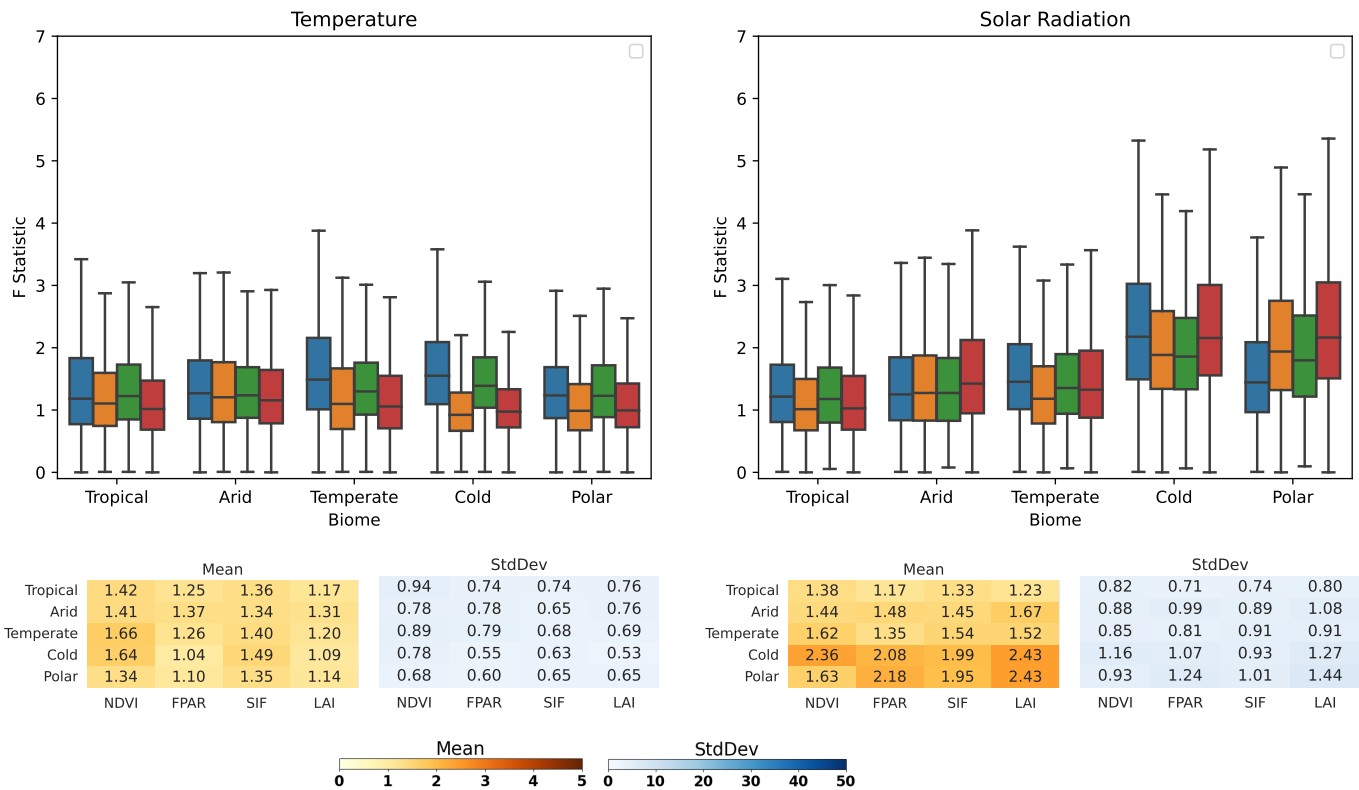

**Figure 5.** Box and whisker plots representing the F values grouped as per the five major Köppen–Geiger biomes. Additional heatmaps show the corresponding mean and standard deviation (StdDev) for each group.

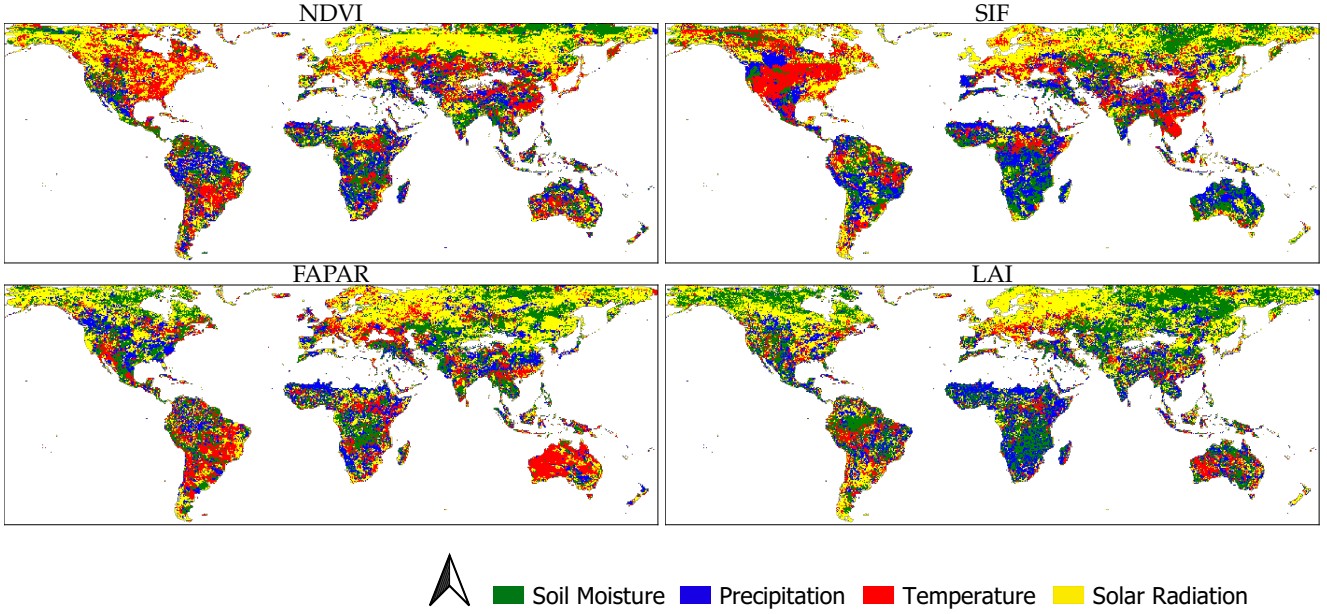

**Figure 6.** Maximum per-pixel F Statistic on a global scale per variable. Each pixel shows the environmental variable that is the dominant driver at the given location.

### 3.3. Main ELV Factors Causing Vegetation Dynamics

Aiming to untangle the most causal environmental variables for each VP variable, a global per-pixel maxima analysis was conducted incorporating all ELVs. Figure 6 depicts the strength of the driving effect of the ELVs compared to each other for different VPs.

Here, for each VP the most causal ELV was displayed on a per-pixel basis. All ELV→VP maps for each VP were superimposed, and the per-pixel maxima of the F stat is mapped.

SM strongly drives VPs over the northernmost latitudes, with regions over Northern Russia (for all VPs) and Northern Canada (the FAPAR and LAI) being the most dependent on SM for all variables. Water availability (SM in conjunction with P) proved to be the strongest causal driver in vast arid areas. SM and P are the main factors for driving vegetation anomalies over the arid areas in the African continent, with P solely being the dominant ELV for causality over the Iberian Peninsula, Northern Australia, and over scattered regions in Northern America for SIF.

Solar radiation tends to be the driving variable over the higher latitudes, especially over Northern Russia, amongst all VPs. The LAI, SIF, and the NDVI are also strongly dependent on R in Northern Europe. Northern latitudes over the American continent (regions in Canada and Alaska) show more heterogeneity considering the primary driver of ELVs.

T shows an even F stat distribution over the globe, consequently being a primary driver ELV on vegetation, whereas other ELVs show no significant causality. As a driving variable, T is dominant for the FAPAR over Australia and in some regions in Southern America, as well as for SIF in North America and Southeast Asia.

## 4. Discussion

The global GC analysis of various VPs opens up new avenues in ecosystem monitoring. Even though multiple studies have presented different analyses on GC, either by using long-term NDVI time series [56,57] or regional scale assessments [40,41], the planetary-scaled, spatially explicit, multiple VP sensitivity analysis from environmental drivers is still lacking. Therefore, we evaluated whether a global GC with four VPs can yield results with sufficient fidelity, by only using two years as input.

### 4.1. Water (SM and P)-Caused Vegetation Anomalies

Having derived global F statistic maps for all ELV→VP links, the results were further subdivided into dominating biomes. In this way, terrestrial vegetation anomalies and sensitivity can be interpreted with additional spatial and climatological details. Throughout the analysis, the LAI expressed the greatest sensitivity for water availability, namely P and SM (mean F stat = 4.05 SM→LAI, mean F stat. = 2.90 P→LAI). The water availability-induced vegetation anomalies over arid biomes have been also confirmed by [58–60]. A possible reason why even small amounts of precipitation and soil moisture increments trigger a prominent positive change in vegetation productivity is the typical response of plant species dominantly inhabiting these (semi-) arid biomes. Variations in the frequency and magnitude of precipitation events can induce changes in the accessibility of water within shallow soil layers, leading to disparate configurations in both the profundity and the temporal extent of soil moisture infiltration [61]. The explanation of the VPs' strong water dependency in arid regions can be found in the literature. Plant genotypes in arid areas are adapted to the high variability in precipitation and show strong responses to pulsed inputs of rainfall events that drive season-long ecosystem dynamics [62,63]. A water availability experiment on the dominant species (*Bouteloua gracilis*) was set up in this region [64]. This study concluded that the plant species showed positive responses: even a small rainfall event (∼5 mm) increased the leaf water potential and stomatal conductance in less than 12 h and lasted for up to two days. Stomatal conductance is a key trait that is strongly associated with leaf carbon budget and photosynthetic activity [65], while leaf water potential is directly correlated to the LAI [66]. This sensitive response by vegetation aligns precisely with the observations made over arid regions, as depicted in the GC temporal profiles in Figure 2.

An interesting phenomenon also occurred throughout the analysis of water-induced vegetation anomalies. Comparing the results of SM and P in Figure 5, a high causality of SM on vegetation over cold and polar biomes can be noticed but to a lesser extent for

P, as was also noted by Shi et al. [67]. The reasoning for the higher causality from SM can be explained by the snow cover. Throughout the melting seasonal snowpacks, soil moisture increases and thus enhances vegetation productivity [68,69]. The high causality on vegetation from snowmelt-induced increments in SM, especially over Northeastern Russia, was also recognized by Grippa et al. [70]. This finding is also confirmed by Figure 4, indicating the higher fractions of areas being explicitly caused on arid, cold, and polar biomes by SM. A noteworthy caveat is the case of the NDVI as compared to other VPs over polar biomes. The FAPAR, SIF, and the LAI causalities cover larger areas of the polar biome (0.3) than the NDVI (0.15). Due to the high spectral albedo of snow, even the lowest fractional snow covers will cause the NDVI measurement to be lower than the value for snow-free conditions [71]. Thus, even though vegetation greening occurs, the red and NIR signatures are suppressed by snow cover, and the satellite's sensor is unable to capture the underlying trends [72]. Using higher-level products such as the FAPAR, the LAI, and SIF bypasses the aforementioned issue.

The negative effects of global changes arising from shifts in the frequency and magnitude of water availability are anticipated to exert influences on the bio-geochemical processes within ecosystems limited by water [73]. Furthermore, the arid ecosystem shall extend in its area across the globe, and it will put increasing pressure on our planet's ecosystem health; more areas will be highly dependent on water, as predicted by Beck et al. [55].

### 4.2. Energy (T and R)-Caused Vegetation Anomalies

The presented results indicate that nearly all VPs are highly sensitive to solar radiation over cold and polar biomes. These results are similar to the findings of Green et al. [74], who investigated the GC links between photosynthetically active radiation (PAR) on SIF. The strong dependency of vegetation over cold and polar biomes, especially in Northern Russia and Northern Scandinavia (see Figure 3), can be explained by the linkage of snowmelt and solar irradiance. The snow cover dominating over higher latitudes typically causes high reflectance within visible to NIR wavelength domains. In contrast, being a near-black body, (cold) snow absorbs and re-emits most thermal infrared radiation [75,76]. This infrared component can melt snow under bright conditions when the temperature is below zero, and, consequently, the increased SM aids photosynthetic activity [77]. Thus, the causality by radiation is higher as opposed to that by temperature over these areas: (e.g., cold biome T→LAI mean F stat = 1.09 versus cold biome R→LAI mean F stat = 2.43). The greater influence of irradiance than the air temperature on snowmelt was also confirmed by Zuzel and Cox [78]. Favorable growing conditions over colder biomes, where the cyclic variations in $CO_2$ fluxes are controlled by light-limited gross primary productivity (GPP), also reveal the underlying driving force from solar radiation [79]. The general tendency for the primary control of R on higher latitudes is also visible in Figure 6, with additional influences from SM. Analogous findings were presented by Papagiannopoulou et al. [57]. The low causality from radiation in tropical areas can be linked to the long convective season caused by the moist atmospheric boundary layer [80]. This results in nearly omnipresent inter-tropical convergence zone cumulus cloud layer clouds blocking incoming solar radiation [81] and hindering the consistency of gap-free data streams observed by satellites [82]. Moreover, the weak response to R over lower latitudes can be further reasoned by the biotic factors controlling canopy photosynthesis [83].

Temperature leads to a relatively even distribution of F stats. Consequently, T seems to drive F stats that are more evenly spatially distributed, thus being primary causal factors over areas, where other factors have no significant effect on VPs, as per Figure 6. Hence, T is the dominant driver over North America, Europe, and Eastern Asia for SIF and over Western Australia for the FAPAR. This spatial distribution agrees with the findings of Marcolla et al. [79]. A stronger T→VP causality over temperate regions can possibly be linked to the higher dependency of vegetation on temperature during growing seasons. Similar results were also presented by Chmielewski and Rötzer [84], Menzel and Fabian [85]

at European continental scales, and by Bao et al. [86], who also confirmed the reliance of the NDVI on T over various climatic zones in China.

*4.3. Limitations and Opportunities for Future Improvements*

The global scale GC analysis of distinct VPs, acquired by multiple spaceborne sensors, is not without its inherent limitations, which must be acknowledged and addressed. The analysis of the two-year time frame already yielded consistent results, whilst still being relatively short, as compared to multi-decadal analysis [87,88]. Using two years of input data showed significant improvements as compared to only using one year. However, when augmenting the temporal window to three years the results showed changes that were not significantly different compared to the two-year analysis. Increasing to multi-decadal temporal domains could possibly yield additional findings; however, the TROPOSIF dataset's availability limited the time frame of analysis. Another limitation induced by the TROPOSIF dataset is its product integrity. At the current state of the art, the valuable physiological information contained in the SIF data remains obscure. This can be attributed to SIF retrieval errors, the SIF signal's weakness compared to reflected radiation, and the high signal-to-noise ratio needed to capture telluric atmospheric absorption lines in the red and far-red bands [89,90].

To gain a deeper understanding of the global biophysical and climatological interconnections, a reversed-directional GC would be the next step, analyzing the vegetation's effect on its surrounding environment. Retrieving GC metrics in the reverse direction (VP→ELV) could potentially unravel additional environmental connections. Expanding the investigation with such results may reveal interconnections in tropical deforestation to reduced evapotranspiration and, thus, precipitation [91] or the influences of root water uptakes on soil moisture [92]. Further advancements can be achieved when calculating the GC that involve various environmental factors. For situations with three or more time series, the extended GC is proposed [93] The case for the GC link between R→VP revealed high causality, but further analyses could reveal the full chain of causal connections: R→snowmelt→SM→VP.

Extending GC analysis to monitor additional factors, such as anthropogenic influences, global forcing from El Niño Southern Oscillation (ENSO), North Atlantic Oscillation (NAO), and influences from regional circulations could enhance the understanding of global interconnections between environment and vegetation [94]. Analyzing additional ELVs, such as vapor pressure deficit (VPD), which is also a driving factor in stomatal opening, carbon uptake, and photosynthetic activity, could further reveal more complex causal links, although VPD is closely coupled to SM and T, and to what extent vegetation productivity in different climate zones is controlled by VPD remains in question [95].

It is expected that future extensions of the GC model with machine learning autoregressive methods, such as neural networks [96] or random forests [97], will enable the modeling of the aforementioned multi-variate non-linear systems. Additionally, varying the lag value of previous observations, at which the VAR model develops the autoregressive model, may also lead to further pinpointing of distinct response characteristics governing the ELV→VP links [34].

Finally, with the upcoming ESA Earth Explorer 8 Fluorescence Explorer (FLEX) mission in mind, the expected full-SIF data streams provided at a 300 m resolution will provide a more accurate insight into terrestrial vegetation health, stress, and resilience [98,99]. Hence, we anticipate that the presented causality analysis will greatly benefit from upcoming FLEX data streams.

## 5. Conclusions

The Granger Causality (GC) method enabled us to investigate vegetation anomalies caused by environmental drivers on a global scale. This study was dedicated to untangling the causality from precipitation, soil moisture, solar radiation, and temperature on four principal vegetation products, i.e., the FAPAR, the LAI, SIF, and the NDVI. Biome-specific

analyses were conducted to evaluate how different ecosystems react to abiotic external factors. The main findings are summarized as follows:

- Water availability (i.e., SM and P) is a strong driver in arid areas, especially for the LAI, which is highly sensitive (0.43 for SM→LAI and 0.41 P→LAI cover a fraction of G-Caused pixel arid biomes).
- SM also causes the LAI on cold and polar biomes with fractions of 0.44 and 0.5, respectively.
- Ecosystems at higher latitudes with cold and polar biomes are driven mainly by R, although R is set to cause the melting of snow, driving soil moisture dynamics. Both on cold and polar biomes, G-Caused areas cover more than 40% of the biomes' areas.
- T causality is evenly distributed amongst all biomes and VPs, with cover fractions of ∼0.1–0.2.

The novelty of the presented analysis is that the role of environmental drivers was investigated in the responses of four fundamentally distinct VPs. The introduced GC method allows for a spatially explicit way to quantify different ecosystem's sensitivity, by only using two years of data. Extending the current investigation with additional multi-decadal temporal data, ELVs or ENSO/NAO teleconnections could reveal further causal effects on vegetation. Furthermore, the upcoming data streams from the FLEX mission are anticipated to mitigate the TROPOSIF-induced uncertainties and to provide a more precise representation of the causal links between variables.

**Author Contributions:** D.D.—Conceptualization, Methodology, Software, Writing—original draft. P.R.-M.—Software—original draft. E.A.—Conceptualization—review and editing. K.B.—Formal analysis, Writing—original draft. J.V.—Formal analysis, Writing—original draft. All authors have read and agreed to the published version of the manuscript.

**Funding:** The research was funded by the European Research Council (ERC) under the FLEXINEL project: grant number 101086622. D.D.K., P.R.M, K.B., and J.V. were funded by the European Union (ERC, FLEXINEL, 101086622). The views and opinions expressed are, however, those of the author(s) only and do not necessarily reflect those of the European Union or the European Research Council. Neither the European Union nor the granting authority can be held responsible for them.

**Data Availability Statement:** The code for the processing of the global GC analysis can be accessed via https://github.com/daviddkovacs/Granger_Causality_global (accessed on 28 August 2023).

**Conflicts of Interest:** The authors declare no conflict of interest.

## Appendix A

### GC p-values 1,2 and 3 years. P->LAI

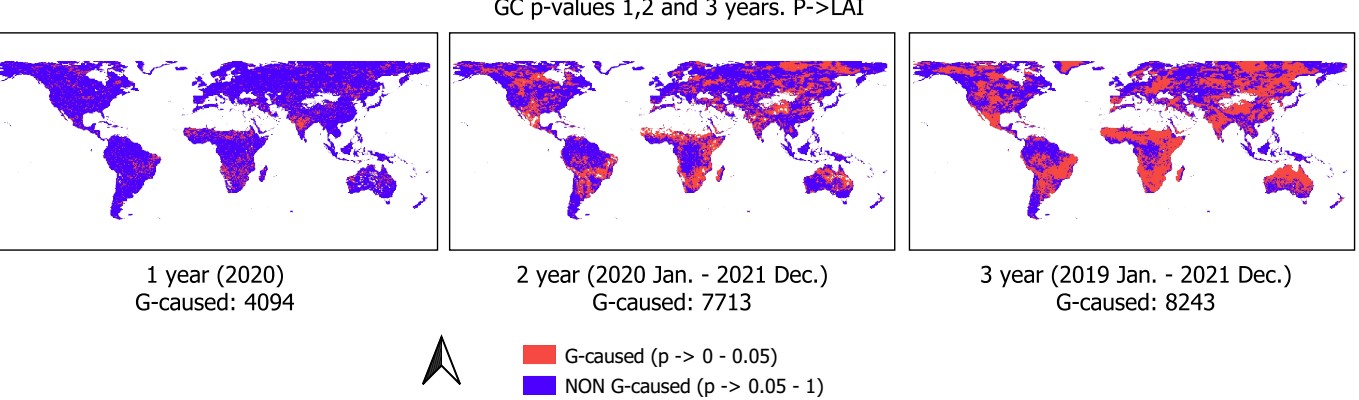

| 1 year (2020) | 2 year (2020 Jan. - 2021 Dec.) | 3 year (2019 Jan. - 2021 Dec.) |
| G-caused: 4094 | G-caused: 7713 | G-caused: 8243 |

G-caused (p -> 0 - 0.05)
NON G-caused (p -> 0.05 - 1)

**Figure A1.** GC *p*-value pixels, investigated during time windows of 1, 2, and 3 years. Explicitly Granger Caused pixel counts are shown below each map, P→LAI.

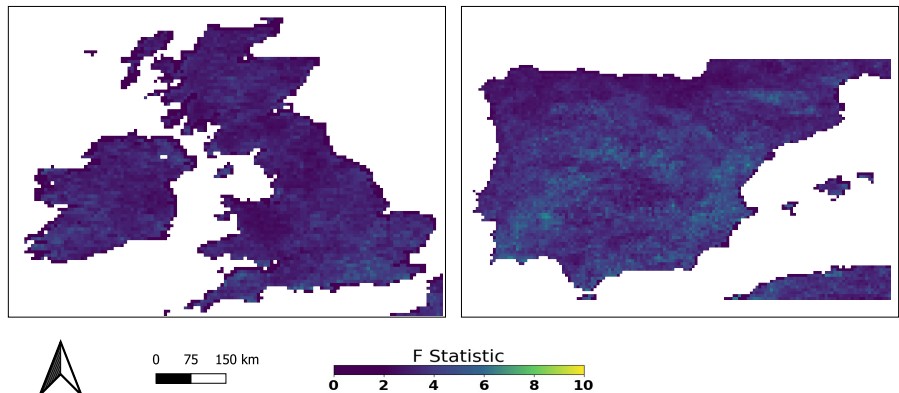

**Figure A2.** Two regional F test maps, P→LAI, exported at 11 km spatial resolution for 2020–2022.

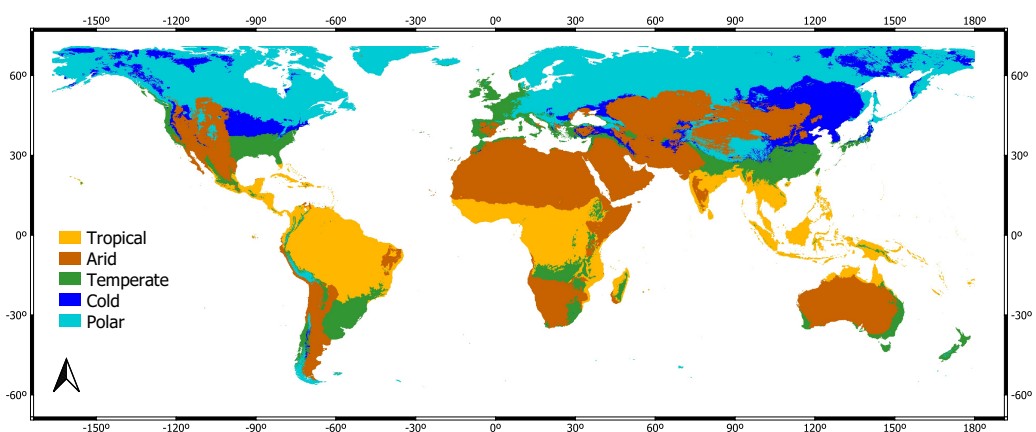

**Figure A3.** Global distribution of Köppen–Geiger biomes categorized into 5 main dominant climates: tropical, arid, temperate, cold, and polar.

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
