# Peer review of "Untangling the Causal Links between Satellite Vegetation Products and Environmental Drivers on a Global Scale by the Granger Causality Method"

_remotesensing, doi:10.3390/rs15204956_

Round 1

Reviewer 1 Report

This paper utilized four distinct satellite vegetation products: FAPAR, LAI, SIF and NDVI, and the granger causality statistical test method to investigate the causal links between satellite vegetation products and environmental stressors on a global scale.

1.     Section introduction: please provide the main research methods in existing studies, what methods others used? Why choose the granger causality statistical test method?

2.     What is the innovation of this study? And what distinguishes this manuscript from other studies?

3.     Please describe clearly all the remote sensing vegetation products utilized in this manuscript, such as their spatial resolution and temporal resolution, the provider, …? How do you reconcile the many distinct products with different spatial resolution and temporal resolution?

4.     In this work, you just use two years dataset (2020 January – 2021 December). Will the results change when a longer period of data are utilized? Why did you just choose these two years of data? Is it possible until December 2022?

Reviewer 2 Report

Major comments:

  1. Page 2: What is meant by vegetation indices to have low repeatability?
  2. Page 2: Water availability drives the regulation of genes. I don’t understand this.
  3. What is meant by a stressor? This needs explanation and more studies to be reported for a better understanding of the readers.
  4. The requirement of normalizing the time series is understood. However, it is not clear why the authors chose their selected method of normalizing (Eqn. 3).
  5. The authors state that the reverse case of Granger Causality i.e. VP-to-ELV was not considered. However, the probability of this causality needs to be checked as well.
  6. One major limitation of this study is not considering the humidity factor explicitly. The authors should also show the GC effect of VPD on the VPs like other considered ELVs. Precipitation and soil moisture do not necessarily account for this.
  7. Page 6: This sentence is not clear where the authors state that the SIF product in the 735-758 nm range offers improved signal-to-noise ratio but remains susceptible to atmospheric influences. This is a contradictory statement in itself.
  8. Page 6: The authors did the analysis at 50 km spatial resolution. This is not very fine and may have implications as the authors wanted to study small-scale interactions between ELVs and VPs.
  9. Page 8: It is strange that boreal regions such as Scandinavia and Northern Russia are controlled by R. This needs further explanation.
  10. Page 13: I disagree with this statement by the authors where they claim that the vegetation in semi-arid, desert zones have shallow roots. In contrast, the plants native to this region, such as Acacia etc. are known to grow longer roots to access deeper water aquifers.
  11. Page 13: The authors make contradictory statements here. How can the snow cover have 98-99% emissivity and still absorb all the infrared radiation?

The quality of the English language used in this paper needs improvement at places. There are syntax and typographical errors. 

Reviewer 3 Report

This was a very interesting and well-written article. A few comments/questions below.

- The authors commented on the chosen timescale for the analysis of two years. They explained that the addition of another year did not significantly change the outcome and large timescales was limited due to the data products available. Can the authors comment a bit more on the less than year observations? The authors noted that the two year analysis significantly improved from the one year results, but what metrics were being used to determine this?

- Analyzing the data for different biomes showed some interesting results. Were there any findings that was anomalous or different from other research results?

-Can the authors provide any additional insight into the lack of causation between SM and LAI (Fig 2 non-causal)? Did this result improve when an additional year was added to the dataset?

Fig. 3 would be better if it were larger

Can the authors expound a bit on Fig. 4 results. For example, the Solar Radiation and NDVI percentage is much higher in polar regions vs the other regions. Is this due to snow/ice coverage?

Author Response

Please see attahcment
